# Chromatic Number of Fuzzy Graphs: Operations, Fuzzy Graph Coloring, and Applications

**Zengtai Gong * and Jing Zhang**

College of Mathematics and Statistics, Northwest Normal University, Lanzhou 730070, China
* Correspondence: gongzt@nwnu.edu.cn; Tel.: +86-931-7971430

**Abstract:** We focus on fuzzy graphs with crisp vertex sets and fuzzy edge sets. This paper introduces a new concept of chromatic number (crisp) for a fuzzy graph $\widetilde{G}(V, \widetilde{E})$. Moreover, we define the operations of cap, join, difference, ring sum, direct product, semiproduct, strong product, and Cartesian product of fuzzy graphs. Furthermore, the exact value or the upper boundary of the chromatic number of these fuzzy graphs is obtained based on the $\alpha$-cuts of $\widetilde{G}$. Finally, two applications of the chromatic number to solve the timetabling problem and the traffic light problem are analyzed.

**Keywords:** fuzzy graph; operations of fuzzy graphs; products of fuzzy graphs; chromatic number

## 1. Introduction

Many real-world problems, such as those in information theory, telecommunications, medical diagnosis, transportation, cluster analysis, traffic networks, and so on, have been mathematically modelled by using graphs $G(V, E)$ since Euler's introduction of graph theory in 1735. Although classical graph theory has been widely used in various aspects, it still has some limitations in dealing with incomplete data and uncertain information. For instance, the traffic congestion at an intersection is due to multiple uncertain factors that include the number of vehicles, cycle length of the traffic light, road conditions, and so on. Therefore, a more appropriate tool is needed to handle these vague phenomena in real-life situations.

Many researchers have studied various types of fuzzy graphs since Zadeh [1] introduced the fuzzy theory in 1965. The fuzzy graphs with crisp vertex sets and fuzzy edge sets, also known as type 1 fuzzy graphs, were originally proposed by Kaufmann [2] in 1973. Furthermore, Rosenfeld [3] introduced type 2 fuzzy graphs with fuzzy vertex sets and fuzzy edge sets. Subsequently, many scholars have made a lot of generalizations about fuzzy graphs. For instance, Mordeson and Peng [4] defined the concept of complementariness in fuzzy graphs and studied some operations on fuzzy graphs in 1994. In the same year, Craine [5] investigated the characteristics of fuzzy interval graphs. With the development of fuzzy graph theory, R.H. Goetschel [6] first described the concept of fuzzy hypergraphs in 1995 and gave an $\alpha$-cut of a fuzzy hypergraph. It was Gong and Wang who investigated the operations of fuzzy and strong fuzzy $r$-uniform hypergraphs, such as Cartesian products, strong products, normal products, lexicographic products, unions, and joining in [7]. Notably, in [8,9], they analyzed the equivalence relationship between fuzzy hypergraphs and fuzzy formal concept analysis and fuzzy information systems so as to use fuzzy hypergraphs to process fuzzy information. Thereafter, Gong and Wang proposed a hesitant fuzzy hypergraph model and investigated some operations and the equivalence relationship of hesitant fuzzy hypergraphs in [10]. In order to better analyze fuzzy graphs, Blue et al. [11] discussed the classification of fuzzy graphs based on the various types of "fuzziness" in graphs, then provided algorithmic solutions to the problems of shortest paths and minimum cuts in fuzzy graphs with examples. The concept of bipolar

fuzzy graphs and strong bipolar fuzzy graphs was initiated by Akram [12], who then investigated some important properties of these fuzzy graphs. Recently, Talal Alhawary [13] defined the direct product, semistrong product, and strong product on type 2 fuzzy graphs. Mathew et al. [14] introduced transitive blocks together with their applications in fuzzy interconnection networks. Furthermore, Binu et al. [15] proposed a connectivity index for a fuzzy graph and its application to human trafficking. For more information on fuzzy graphs, readers can refer to [16–21].

Graph coloring is one of the most studied problems in combinatorial optimization. The classical graph-coloring problem is to group objects into as few groups as possible, subject to the constraint that no incompatible objects end up in the same group. The smallest number of groups that is needed to color a graph is called the chromatic number. Many problems in practical situations can be modeled as coloring problems, which are known to be NP-hard [22]. The applications of the graph-coloring problem include wiring problems [23], printed circuits [24], frequency assignment problems [25], resource allocation [26], a wide variety of scheduling problems [27–30], computer register allocation [31], and so on.

Many researchers have experimented with various graph colorings. In some cases, ambiguity is used in coloring problems of fuzzy graphs when dealing with uncertainties such as vagueness. In the year 2005, the process of coloring the fuzzy graphs was implemented by Muñoz [32], who initiated two methods by which to color the vertices of the fuzzy graphs with a crisp vertex set and a fuzzy edge set (the type 1 fuzzy graphs) by means of $\alpha$-cut coloring of the graphs with $\alpha \in [0,1]$ and an extension of distance defined between colors. In that paper, Muñoz et al. also discussed the application of $\alpha$-cut chromatic numbers in arranging traffic flows at an intersection and the application of the second coloring method in timetabling problems. Furthermore, Eslahchi and Onagh [33] constructed this method for fuzzy graphs with fuzzy vertex sets and fuzzy edge sets (the type 2 fuzzy graphs) based on strong adjacencies between vertices. In a similar way, Kishore and Sunitha [34] investigated the strong coloring and chromatic number of type 2 fuzzy graphs based on strong adjacencies between vertices. They also demonstrated how to use strong coloring on fuzzy graphs to solve traffic light problems. Cioban [35] then proposed a method for vertex coloring of type 2 fuzzy graphs by using delta-fuzzy independent vertex sets with $\delta \in [0,1]$, and the chromatic number was dubbed the $\delta$-chromatic number. Recently, an application of vertex coloring on these graphs to solve a cell site assignment problem in a telecommunications network was given by Keshavarz [36]. Moreover, Anjaly Kishore et al. [37] discussed the chromatic number of the resultant type 2 fuzzy graphs in 2016. In the same year, a new concept of coloring fuzzy graphs was introduced by Sovan Samanta et al. in [38]. Sovan Samanta also described some concepts, such as the strength cut graph and the chromatic number of fuzzy graphs.

Even though type 2 fuzzy graphs have been widely used in many research fields, it is more adaptable and perfect to use type 1 fuzzy graphs for some real-life problems. Susana Muñoz et al. [32], for example, used the chromatic number of a type 1 fuzzy graph to analyze traffic lights and a timing problem. Rosyida et al. [39] proposed a new approach to determining the fuzzy chromatic set of fuzzy graphs and verified that the fuzzy chromatic number of fuzzy graphs is a discrete fuzzy number. In that paper, an application of the fuzzy chromatic number of fuzzy graphs to the problem of medical virus infection at some locations was given. Subsequently, Rosyida offered an application of the fuzzy chromatic number to determine the number of phases of an integrated traffic light system in 2020 [40]. Due to the practicability of the type 1 fuzzy graph in real-life situations, we will continue to study this type of fuzzy graph. To the best of our knowledge, the operations on type 1 fuzzy graphs and the relationship between the chromatic numbers of fuzzy graphs and the resultant fuzzy graphs obtained by performing various operations on fuzzy graphs have not been investigated until now.

The structure of this article is as follows. Section 2 discusses some basic concepts in fuzzy set theory and the operations on fuzzy graphs, and a new definition of the chromatic

number of fuzzy graphs is given. In Section 3, we investigate the relationship between the chromatic numbers of fuzzy graphs and that of the resultant fuzzy graphs obtained by performing various operations on fuzzy graphs such as union, cap, join, difference, ring sum, and different types of products. In Section 4, an application example based on the coloring of fuzzy graphs with crisp vertex sets and fuzzy edge sets is provided. Finally, conclusions are given in Section 5. We discuss some terminologies in fuzzy graphs and fuzzy graph coloring, as cited in [32,39,40].

## 2. Preliminaries

In this section, we first recall some definitions in graph theory, fuzzy set theory, and fuzzy graph coloring that will be used in constructing chromatic numbers of fuzzy graphs. Then the definitions of operations on fuzzy graphs are introduced.

### 2.1. k-Coloring of Crisp Graph

A classical graph $G(V, E)$ consists of a nonempty vertex set $V = V(G)$ and an edge set $E = E(G)$. An edge $(u, v)$ is said to be incident to the vertices $u$ and $v$. Two vertices linked by an edge are called adjacent. Furthermore, we call classical graph $G(V, E)$ as a crisp graph.

For a crisp graph $G(V, E)$, there are two approaches to defining the $k$-coloring. In the first approach, a proper $k$-coloring of a graph $G(V, E)$ with vertex set $V(G)$ and edge set $E(G)$ is a mapping $f$ from $V$ to the set $\{1, 2, \cdots, k\}$ such that $f(u) \neq f(v)$ whenever $(u, v)$ is an edge in $E$. The chromatic number of $G$ is the smallest $k$ for which $G$ admits a proper $k$-coloring, written by $\chi(G)$. In the second approach, a $k$-coloring of graph $G(V, E)$ is viewed as a partition of $\{S_1, S_2, \cdots, S_k\}$ of $V$, where $S_i$ denotes the (possibly empty) set of vertices assigned colour $i$ ($i = 1, 2, \cdots, k$), $S_i \cap S_j = \varnothing$ for $i \neq j$, and $S_1 \cup S_2 \cup \cdots \cup S_k = V$. A graph is $k$-colourable if it has a $k$-coloring. The minimum $k$ for which a graph $G$ is $k$-colourable is called its chromatic number.

In the year 2007, Cioban [35] generalized the concept of $k$-coloring of crisp graph $G$ in fuzzy graphs by partitioning of the vertex set of $G$ into independent vertex sets. A vertex subset $S \subseteq V$ is described as an independent vertex set of $G(V, E)$ if $(u, v) \notin E$ for all $u, v \in S$.

### 2.2. Basic Concepts in Fuzzy Sets

**Definition 1** ([1])**.** *Let $X$ be a nonempty universal set. A fuzzy set $\widetilde{A}$ on $X$ is defined as a mapping $\mu_{\widetilde{A}} : X \to [0, 1]$, where $\mu_{\widetilde{A}}(x)$ is the membership degree of $x$ to the fuzzy set $\widetilde{A}$. We denote by $\widetilde{F}(X)$ the collection of all fuzzy subsets of $X$.*

Fuzzy sets are generalizations of the classical sets represented by their characteristic functions $\mu_{\widetilde{A}} : X \to \{0, 1\}$. In our case $\widetilde{A}(x) = 1$ means full membership of $x$ in $\widetilde{A}$, whereas $\widetilde{A}(x) = 0$ expresses nonmembership, but in contrary to the classical case other membership degrees are allowed. A set $h(\widetilde{A}) = \sup\{\mu_{\widetilde{A}}(x) | x \in X\}$ is called a height of fuzzy set $\widetilde{A}$. Moreover, a fuzzy set $\widetilde{A}$ is said to be a normal fuzzy set if $h(\widetilde{A}) = 1$.

**Definition 2** ([1])**.** *Let $\widetilde{A} : X \to [0, 1]$ be a fuzzy set. The level sets of $\widetilde{A}$ are defined as the classical sets*

$$A_\alpha = \{x \in X | \widetilde{A}(x) \geq \alpha\},$$

$0 < \alpha \leq 1$.

$$A_1 = \{x \in X | \widetilde{A}(x) = 1\}$$

*is called the core of the fuzzy set $\widetilde{A}$, and*

$$\operatorname{supp}\widetilde{A} = \{x \in X | \widetilde{A}(x) > 0\}$$

*is called the support of the fuzzy set.*

**Definition 3** ([1]). *Let $\sigma$ be a fuzzy subset of a set S and $\mu$ be a fuzzy relation on S. Then $\mu$ is called a fuzzy relation on $\sigma$ if*

$$\mu(x,y) \leq \sigma(x) \wedge \sigma(y), \text{ for all } x, y \in S.$$

*A fuzzy relation $\mu$ is symmetric if $\mu(x,y) = \mu(y,x)$ for all $x,y \in X$.*

**Definition 4** ([1]). *Let $\widetilde{A}, \widetilde{B} \in \widetilde{F}(X)$. We say that the fuzzy set $\widetilde{A}$ is included in $\widetilde{B}$ if*

$$\widetilde{A}(x) \leq \widetilde{B}(x), \text{ for all } x \in X.$$

*We denote $\widetilde{A} \subseteq \widetilde{B}$.*

**Definition 5** ([1]). *Let $\widetilde{A}, \widetilde{B} \in \widetilde{F}(X)$. The intersection of $\widetilde{A}$ and $\widetilde{B}$ is the fuzzy set $\widetilde{C}$ with*

$$\widetilde{C}(x) = \min\{\widetilde{A}(x), \widetilde{B}(x)\} = \widetilde{A}(x) \wedge \widetilde{B}(x), \text{ for all } x \in X.$$

*We denote $\widetilde{C} = \widetilde{A} \cap \widetilde{B}$.*

**Definition 6** ([1]). *Let $\widetilde{A}, \widetilde{B} \in \widetilde{F}(X)$. The union of $\widetilde{A}$ and $\widetilde{B}$ is the fuzzy set $\widetilde{C}$, where*

$$\widetilde{C}(x) = \max\{\widetilde{A}(x), \widetilde{B}(x)\} = \widetilde{A}(x) \vee \widetilde{B}(x), \text{ for all } x \in X.$$

*We denote $\widetilde{C} = \widetilde{A} \cup \widetilde{B}$.*

**Definition 7** ([1]). *Let $\widetilde{A} \in \widetilde{F}(X)$ be a fuzzy set. The complement of $\widetilde{A}$ is the fuzzy set $\widetilde{B}$ where*

$$\widetilde{B}(x) = 1 - \widetilde{A}(x), \text{ for all } x \in X.$$

*We denote $\widetilde{B} = \bar{\widetilde{A}}$.*

*2.3. Basic Concepts of Fuzzy Graph Coloring*

Kaufmann proposed the concept of fuzzy graphs $\widetilde{G}(V, \widetilde{E})$. Meanwhile, Rosyida initiated the concept of fuzzy graphs $\widetilde{G}(\widetilde{V}, \widetilde{E})$ based on the fuzzy relation.

In this paper, we consider fuzzy graphs $\widetilde{G}(V, \widetilde{E})$, which denote the incompatibility graph. Every fuzzy graph is assumed to be finite, with no loops and an undirected graph.

**Definition 8** ([2]). *Let $G(V, E)$ be a fuzzy graph, where V is the vertex set, the fuzzy edge set $\tilde{E}$ is characterized by the matrix $\mu = \mu(u,v)_{u,v \in V}$: $\mu(u,v) = \mu_{\tilde{E}}(u,v)$, for all $u, v \in V$ such that $u \neq v$, and $\mu_{\tilde{E}}: V \times V \to I$ is the membership function.*

This type of fuzzy graph is also called a type 1 fuzzy graph.

For the sake of brevity, we declare that in this paper, each element $\mu(u,v) \in \widetilde{E}$ represents the incompatible level of the edge $(u,v)$ for all $u, v \in V$ with $u \neq v$. $\mu$, which is chosen suitably in all the examples, is symmetric (i.e., $\mu(u,v) = \mu(v,u)$, for any $(u,v) \in \widetilde{E}$). In addition, we denote the underlying crisp graph of $\widetilde{G}$ by $G^*(V, E^*)$, where $E^* = \{(u,v) \in V \times V | \mu(u,v) > 0, \text{ for all } u, v \in V\}$.

Assume $I = A \cup \{0,1\}$, where $A = \{\alpha_1 \leq \alpha_2 \leq \cdots \leq \alpha_k\}$ is the fundamental set (level set) of $\widetilde{G}$. In fact, a fuzzy graph $\widetilde{G}(V, \widetilde{E})$ can be regarded as a generalization of an incompatibility graph $G(V, E)$. Taking $I = \{0,1\}$, $\widetilde{G}(V, \widetilde{E})$ becomes a crisp graph if matrix $\mu$ is defined as

$$\mu(u,v) = \begin{cases} 1, & \text{if } (u,v) \in \widetilde{E} \\ 0, & \text{otherwise} \end{cases}$$

for any $u, v \in V$.

**Definition 9** ([2]). *Let $\widetilde{G}_1(V_1, \widetilde{E}_1)$ be a fuzzy graph having a crisp vertex set $V_1$ and fuzzy edge set $\widetilde{E}_1$ with membership function $\mu_{\widetilde{E}_1} : V_1 \times V_1 \to [0, 1]$. The fuzzy graph $\widetilde{G}_1$ is called a fuzzy subgraph of $\widetilde{G}$, if $V_1 \subseteq V$ and $\widetilde{E}_1 \subseteq \widetilde{E}$. Similarly, the fuzzy graph $\widetilde{H}(P, \widetilde{E}_H)$ is called a fuzzy subgraph of $\widetilde{G}$ induced by $P$ if $P \subseteq V$ and $\mu_{\widetilde{E}_H}(u, v) = \mu(u, v)$ for $u, v \in P$. We write $\langle P \rangle$ to denote the fuzzy subgraph induced by $P$.*

The concept of chromatic number of fuzzy graphs (Definition 10) was introduced by Muñoz et al. [32]. The authors considered fuzzy graphs with a crisp vertex set, and edges with membership degree in $[0, 1]$; moreover, they showed that the fuzzy chromatic number of a fuzzy graph is a normalized fuzzy number whose modal value is associated with the empty edge-set graph. Its meaning depends on the sense of index $\alpha$, and it can be interpreted in the following way. For lower values of $\alpha$ there are many incompatible links between vertices and, consequently, more colors are needed in order to consider these incompatibilities. On the other hand, for higher values of $\alpha$ there are fewer incompatible links between vertices and fewer colors are needed (see [32]). In order to deal with the ambiguity problem, the chromatic number summarizes all this information.

Given a fuzzy graph $\widetilde{G}(V, \widetilde{E})$, which can be characterized by the family of its $\alpha$-cut graphs, and the membership function of $\widetilde{E}$ is $\mu$. In order to obtain more information about it, a natural approach is to analyze the family of the $\alpha$-cut graphs. Let $G_\alpha = \{(V, E_\alpha) | \alpha \in I\}$ be the family of $\alpha$-cut sets of $\widetilde{G}$, and the $\alpha$-cut of a fuzzy graph be a crisp graph $G_\alpha(V, E_\alpha)$ with $E_\alpha = \{(u, v) | u, v \in V, \mu(u, v) \geq \alpha\}$. Hence, any (crisp) $k$-coloring can be defined on $G_\alpha$. The $k$-coloring function of $\widetilde{G}$ is defined through this sequence. For each $\alpha \in I$, let $\chi_\alpha$ denote the chromatic number of $G_\alpha$. The chromatic number of $\widetilde{G}$ is defined through a monotone family of sets [32].

**Definition 10** ([32]). *If $\widetilde{G}(V, \widetilde{E})$ is such a fuzzy graph, where $V = \{1, 2, 3, \cdots, n\}$ and $\mu$ is a fuzzy number on the set of all subsets of $V \times V$. Assume $I = A \cup \{0, 1\}$, where $A = \{\alpha_1 \leq \alpha_2 \leq \cdots \leq \alpha_k\}$ is the fundamental set (level set) of $\widetilde{G}$. For each $\alpha_i \in I$, $G_{\alpha_i}$ denotes the crisp graph $G_{\alpha_i}(V, E_{\alpha_i})$, where $E_{\alpha_i} = \{(u, v) | \mu(u, v) \geq \alpha_i\}$ and $\chi_{\alpha_i} = \chi(G_{\alpha_i})$ denotes the chromatic number of crisp graph $G_{\alpha_i}$.*

Based on Definition 10, we extend the chromatic number of fuzzy graph $\widetilde{G}(V, \widetilde{E})$ as the max number of $\chi(G_{\alpha_i})$ as follows.

**Definition 11.** *Let $\widetilde{G}(V, \widetilde{E})$ be a fuzzy graph, $\mu$ be a fuzzy number on the set of all subsets of $V \times V$, and $A = \{\alpha_1, \alpha_2, \cdots, \alpha_k\}$ be the fundamental set of $\widetilde{G}$. For any $\alpha_i \in A$, $G_{\alpha_i}$ denotes the crisp graph $G_{\alpha_i}(V, E_{\alpha i})$ with $E_{\alpha i} = \{(u, v) | \mu(u, v) \geq \alpha_i\}$, and $\chi(G_{\alpha_i})$ denotes the chromatic number of the crisp graph $G_{\alpha_i}$. The chromatic number of fuzzy graph $\widetilde{G}$ is the number $\chi(\widetilde{G}) = \max\{\chi(G_{\alpha_i}) | \alpha_i \in A\}$.*

**Definition 12** ([40]). *(Union) Let $V_1$ and $V_2$ be finite nonempty sets, $\widetilde{G}_1(V_1, \widetilde{E}_1)$ and $\widetilde{G}_2(V_2, \widetilde{E}_2)$ be two fuzzy graphs where the membership functions of $\widetilde{E}_1$ and $\widetilde{E}_2$ are $\mu_1$ and $\mu_2$, respectively. A union of $\widetilde{G}_1$ and $\widetilde{G}_2$ is a fuzzy graph $\widetilde{G}(V, \widetilde{E}) = \widetilde{G}_1 \cup \widetilde{G}_2$ which has a vertex set $V = V_1 \cup V_2$ and a fuzzy edge set $\widetilde{E} = \widetilde{E}_1 \cup \widetilde{E}_2$ with $\mu_{\widetilde{E}}(u, v) = \max\{\mu_1(u, v), \mu_2(u, v)\}$, for all $u, v \in V$. If $V_1 \cap V_2 = \varnothing$, then $\widetilde{G}(V, \widetilde{E}) = \widetilde{G}_1 \cup \widetilde{G}_2$ is called a disjointed union of fuzzy graphs $\widetilde{G}_1$ and $\widetilde{G}_2$.*

*2.4. Operations on Fuzzy Graphs*

**Definition 13.** *(Isomorphic) Let $\widetilde{G}_1(V_1, \widetilde{E}_1)$ and $\widetilde{G}_2(V_2, \widetilde{E}_2)$ be two fuzzy graphs, $\mu_1$ and $\mu_2$ be the membership functions of $\widetilde{E}_1$ and $\widetilde{E}_2$, respectively. We will say that $\widetilde{G}_1$ and $\widetilde{G}_2$ are isomorphic, written by $\widetilde{G}_1 \cong \widetilde{G}_2$, if there exists a bijection $h : V_1 \to V_2$ such that $\mu_1(u, v) = \mu_2(h(u), h(v))$ for all $u, v \in V_1$.*

**Definition 14.** *(Complement) Let $\widetilde{G}(V,\widetilde{E})$ be a fuzzy graph, $\mu$ be the membership functions of $\widetilde{E}$. The complement graph of $\widetilde{G}(V,\widetilde{E})$ is a fuzzy graph $\overline{\widetilde{G}}(V,\overline{\widetilde{E}})$ with $\overline{\mu} = \left(\overline{\mu}(u,v)\right)$, $\overline{\mu}(u,v) = 1 - \mu(u,v)$ for any $(u,v) \in \widetilde{E}$. Moreover, if $\widetilde{G} \cong \overline{\widetilde{G}}$, then $\widetilde{G}$ is called self-complementary.*

**Example 1.** *Let $\widetilde{G}(V,\widetilde{E})$ be a fuzzy graph, $V = \{u_1, u_2, u_3, u_4, u_5\}$, the membership function of $\widetilde{E}$ be*

$$\mu = \begin{bmatrix} - & 0 & 0.6 & 0.2 & 0 \\ 0 & - & 0 & 0.8 & 0.4 \\ 0.6 & 0 & - & 0 & 0.7 \\ 0.2 & 0.8 & 0 & - & 0 \\ 0 & 0.4 & 0.7 & 0 & - \end{bmatrix}.$$

*$\widetilde{G}$ is shown as (a) in Figure 1. For simplicity, we show only the computing of $\overline{\mu}(u_1, u_3)$ and $\overline{\mu}(u_1, u_4)$. By Definition 14, we have $\overline{\mu}(u_1, u_3) = 1 - \mu(u_1, u_3) = 0.4$, $\overline{\mu}(u_1, u_4) = 1 - \mu(u_1, u_4) = 0.8$. The position of $\overline{\mu}(u_1, u_3)$ in matrix $\overline{\mu}$ is the first row and the third column, the position of $\overline{\mu}(u_1, u_4)$ in matrix $\mu$ is the first row and the fourth column. By computing, it is easy to obtain $\overline{\widetilde{G}}(V, \overline{\widetilde{E}})$, as shown as (b) in Figure 1, $V = \{u_1, u_2, u_3, u_4, u_5\}$, and the membership function of $\overline{\widetilde{E}}$ is*

$$\overline{\mu} = \begin{bmatrix} - & 1 & 0.4 & 0.8 & 1 \\ 1 & - & 1 & 0.2 & 0.6 \\ 0.4 & 1 & - & 1 & 0.3 \\ 0.8 & 0.2 & 1 & - & 1 \\ 1 & 0.6 & 0.3 & 1 & - \end{bmatrix}.$$

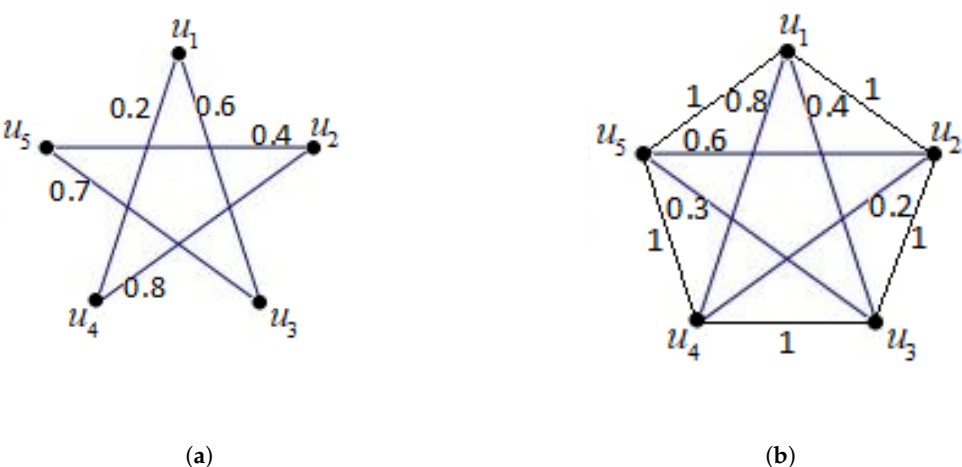

**(a)** **(b)**

**Figure 1.** (a) $\widetilde{G}$ and (b) $\overline{\widetilde{G}}$.

**Definition 15.** *(Fuzzy neighbour edge set) Let $\widetilde{G}(V,\widetilde{E})$ be a fuzzy graph. For a given vertex $u$ in vertex set $V$, the fuzzy edge set $\{(u, u_i) | \mu(u, u_i) > 0, \text{ for all } u_i \in V\}$ is called the fuzzy neighbour edge set of vertex $u$, denoted by $N_{\widetilde{E}(u)}$.*

**Definition 16.** *(Fuzzy complete graph) A fuzzy graph $\widetilde{G}(V,\widetilde{E})$ is a fuzzy complete graph, if $\mu(u,v) > 0$, for all $u, v \in V$.*

**Definition 17** ((Clique) [41])**.** *A clique of simple crisp graphs $G^*$ is a subset $S$ of $V$, such that the graph induced by $S$ is complete.*

**Remark 1.** *Let $\widetilde{G}(V,\widetilde{E})$ be a fuzzy graph with the underlying graph $G^*(V, E^*)$. A clique with maximum number of vertices in $\widetilde{G}$ is a maximal clique of the graph $G^*$.*

**Definition 18.** *(Cap) Let $\widetilde{G}_1(V_1, \widetilde{E}_1)$ and $\widetilde{G}_2(V_2, \widetilde{E}_2)$ be two fuzzy graphs, $\mu_1$ and $\mu_2$ be the membership functions of $\widetilde{E}_1$ and $\widetilde{E}_2$, respectively. The cap of $\widetilde{G}_1$ and $\widetilde{G}_2$ is a fuzzy graph $\widetilde{G}(V, \widetilde{E}) = \widetilde{G}_1 \cap \widetilde{G}_2$ which has a fuzzy edge set $\widetilde{E} = \widetilde{E}_1 \cap \widetilde{E}_2$, and $\mu(u, v) = \min\{\mu_1(u, v), \mu_2(u, v)\}$, for all $(u, v) \in \widetilde{E}$.*

**Definition 19.** *(Difference) Let $\widetilde{G}_1(V_1, \widetilde{E}_1)$ and $\widetilde{G}_2(V_2, \widetilde{E}_2)$ be two fuzzy graphs, $\mu_1$ and $\mu_2$ be the membership functions of $\widetilde{E}_1$ and $\widetilde{E}_2$, respectively. The difference of $\widetilde{G}_1$ and $\widetilde{G}_2$ is a fuzzy graph $\widetilde{G}(V, \widetilde{E}) = \widetilde{G}_1 - \widetilde{G}_2$ which has a fuzzy edge set $\widetilde{E} = \widetilde{E}_1 - \widetilde{E}_2 = \{(u, v)|(u, v) \in \widetilde{E}_1 \text{ and } (u, v) \notin \widetilde{E}_2\}$, and $\mu(u, v) = \mu_1(u, v) \wedge (1 - \mu_2(u, v))$, for all $u, v \in V_1 \cup V_2$.*

**Definition 20.** *(Ring sum) Let $\widetilde{G}_1(V_1, \widetilde{E}_1)$ and $\widetilde{G}_2(V_2, \widetilde{E}_2)$ be two fuzzy graphs, $\mu_1$ and $\mu_2$ be the membership functions of $\widetilde{E}_1$ and $\widetilde{E}_2$, respectively. The ring sum of $\widetilde{G}_1$ and $\widetilde{G}_2$ is a fuzzy graph*

$$\widetilde{G}(V, \widetilde{E}) = \widetilde{G}_1 \oplus \widetilde{G}_2 = (\widetilde{G}_1 \cup \widetilde{G}_2) - (\widetilde{G}_1 \cap \widetilde{G}_2)$$
$$= (\widetilde{G}_1 - \widetilde{G}_2) \cup (\widetilde{G}_2 - \widetilde{G}_1).$$

**Remark 2.** *The ring sum of two fuzzy graphs satisfies commutative law and associative law, i.e.,*

$$\widetilde{G}_1 \oplus \widetilde{G}_2 = \widetilde{G}_2 \oplus \widetilde{G}_1$$
$$\widetilde{G}_1 \oplus (\widetilde{G}_2 \oplus \widetilde{G}_3) = (\widetilde{G}_1 \oplus \widetilde{G}_2) \oplus \widetilde{G}_3.$$

**Definition 21.** *(Join) Let $\widetilde{G}_1(V_1, \widetilde{E}_1)$ and $\widetilde{G}_2(V_2, \widetilde{E}_2)$ be two fuzzy graphs, $\mu_1$ and $\mu_2$ be the membership functions of $\widetilde{E}_1$ and $\widetilde{E}_2$, respectively. The join of two disjoint fuzzy graphs $\widetilde{G}_1$ and $\widetilde{G}_2$ is a fuzzy graph $\widetilde{G}(V, \widetilde{E}) = \widetilde{G}_1 + \widetilde{G}_2$, where $V = V_1 \cup V_2$, $\widetilde{E} = \widetilde{E}_1 \cup \widetilde{E}_2 \cup \widetilde{E}'$, $\widetilde{E}'$ is the set of all edges joining $V_1$ and $V_2$, and the membership function of $\widetilde{E}$ is $\mu = \left(\mu(u, v)\right)$, $\mu(u, v)$ is calculated as follows:*

$$\mu(u, v) = \begin{cases} \mu_1(u, v), & (u, v) \in \widetilde{E}_1, \\ \mu_2(u, v), & (u, v) \in \widetilde{E}_2, \\ \min\{\mu_1(N_{\widetilde{E}_1}(u)), \mu_2(N_{\widetilde{E}_2}(v))\}, (u \in V_1, v \in V_2), & (u, v) \in \widetilde{E}'. \end{cases}$$

**Example 2.** *Given two fuzzy graphs $\widetilde{G}_1(V_1, \widetilde{E}_1)$ and $\widetilde{G}_2(V_2, \widetilde{E}_2)$, where $V_1 = \{u_1, u_2, u_3\}$ and $V_2 = \{v_1, v_2\}$, the membership functions of $\widetilde{E}_1$ and $\widetilde{E}_2$ are $\mu_1$ and $\mu_2$, respectively:*

$$\mu_1 = \begin{bmatrix} - & 0.4 & 0 \\ 0.4 & - & 0.8 \\ 0 & 0.8 & - \end{bmatrix}, \qquad \mu_2 = \begin{bmatrix} - & 0.7 \\ 0.7 & - \end{bmatrix}.$$

*By Definition 21, we can obtain the join of fuzzy graphs $\widetilde{G}_1$ and $\widetilde{G}_2$, denoted by $\widetilde{G}(V, \widetilde{E})$, where $V = \{u_1, u_2, u_3, v_1, v_2\}$ and the membership function of $\widetilde{E}$ is $\mu$ as follows.*

$$\mu = \begin{bmatrix} - & 0.4 & 0 & 0.4 & 0.4 \\ 0.4 & - & 0.8 & 0.4 & 0.4 \\ 0 & 0.8 & - & 0.7 & 0.7 \\ 0.4 & 0.4 & 0.7 & - & 0.7 \\ 0.4 & 0.4 & 0.7 & 0.7 & - \end{bmatrix}.$$

*For simplicity, we show only the computing of $\mu(u_1, v_1)$ and $\mu(u_3, v_1)$. By Definition 21, we have*

$$\mu(u_1, v_1) = \min\{\mu_1(N_{\widetilde{E}_1}(u_1)), \mu_2(N_{\widetilde{E}_2}(v_1))\} = 0.4,$$

$$\mu(u_3, v_1) = \min\{\mu_1(N_{\widetilde{E}_1}(u_3)), \mu_2(N_{\widetilde{E}_2}(v_1))\} = 0.7.$$

*The position of $\mu(u_1, v_1)$ in matrix $\mu$ is the fourth row and the first column, the position of $\mu(u_3, v_1)$ in matrix $\mu$ is the fourth row and the third column.*

*Fuzzy graphs $\widetilde{G}_1(V_1, \widetilde{E}_1)$, $\widetilde{G}_2(V_2, \widetilde{E}_2)$ and $\widetilde{G}(V, \widetilde{E})$ are shown as (a), (b), and (c) in Figure 2, respectively.*

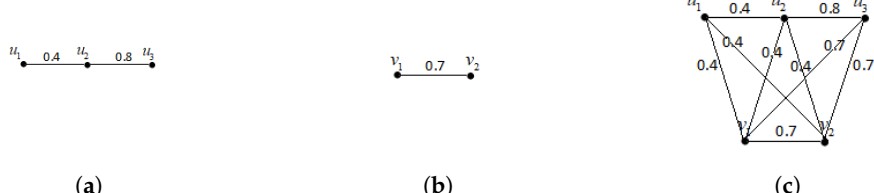

(a)               (b)               (c)

**Figure 2.** (a) $\widetilde{G}_1$, (b) $\widetilde{G}_2$, and (c) $\widetilde{G}_1 + \widetilde{G}_2$.

**Definition 22.** *(Direct product) Let $\widetilde{G}_1(V_1, \widetilde{E}_1)$ and $\widetilde{G}_2(V_2, \widetilde{E}_2)$ be two fuzzy graphs, $\mu_1$ and $\mu_2$ be the membership functions of $\widetilde{E}_1$ and $\widetilde{E}_2$, respectively. The direct product of two fuzzy graphs $\widetilde{G}_1$ and $\widetilde{G}_2$ is a fuzzy graph $\widetilde{G}(V, \widetilde{E}) = \widetilde{G}_1 \sqcap \widetilde{G}_2$, where $V = V_1 \times V_2$, and $\widetilde{E} = \{(u_1v_1, u_2v_2) | (u_1, u_2) \in \widetilde{E}_1, (v_1, v_2) \in \widetilde{E}_2\}$,*

$$\mu(u_1v_1, u_2v_2) = \mu_1(u_1, u_2) \wedge \mu_2(v_1, v_2) = \min\{\mu_1(u_1, u_2), \mu_2(v_1, v_2)\}.$$

**Definition 23.** *(Semiproduct) Let $\widetilde{G}_1(V_1, \widetilde{E}_1)$ and $\widetilde{G}_2(V_2, \widetilde{E}_2)$ be two fuzzy graphs, $\mu_1$ and $\mu_2$ be the membership functions of $\widetilde{E}_1$ and $\widetilde{E}_2$, respectively. The semiproduct of two fuzzy graphs $\widetilde{G}_1$ and $\widetilde{G}_2$ is a fuzzy graph $\widetilde{G}(V, \widetilde{E}) = \widetilde{G}_1 \Diamond \widetilde{G}_2$, where $V = V_1 \times V_2$, and $\widetilde{E} = \{(uv_1, uv_2) | u \in V_1, (v_1, v_2) \in \widetilde{E}_2\} \cup \{(u_1v_1, u_2v_2) | (u_1, u_2) \in \widetilde{E}_1, (v_1, v_2) \in \widetilde{E}_2\}$,*

$$\mu(uv_1, uv_2) = \mu_2(v_1, v_2), \mu(u_1v_1, u_2v_2) = \mu_1(u_1, u_2) \wedge \mu_2(v_1, v_2).$$

**Definition 24.** *(Strong product) Let $\widetilde{G}_1(V_1, \widetilde{E}_1)$ and $\widetilde{G}_2(V_2, \widetilde{E}_2)$ be two fuzzy graphs, $\mu_1$ and $\mu_2$ be the membership functions of $\widetilde{E}_1$ and $\widetilde{E}_2$, respectively. The strong product of two fuzzy graphs $\widetilde{G}_1$ and $\widetilde{G}_2$ is a fuzzy graph $\widetilde{G}(V, \widetilde{E}) = \widetilde{G}_1 \otimes \widetilde{G}_2$, where $V = V_1 \times V_2$, and $\widetilde{E} = \{(uv_1, uv_2) | u \in V_1, (v_1, v_2) \in \widetilde{E}_2\} \cup \{(u_1w, u_2w) | (u_1, u_2) \in \widetilde{E}_1, w \in V_2\} \cup \{(u_1v_1, u_2v_2) | (u_1, u_2) \in \widetilde{E}_1, (v_1, v_2) \in \widetilde{E}_2\}$,*

$$\mu(uv_1, uv_2) = \mu_2(v_1, v_2), \mu(u_1w, u_2w) = \mu_1(u_1, u_2), \mu(u_1v_1, u_2v_2) = \mu_1(u_1, u_2) \wedge \mu_2(v_1, v_2).$$

**Definition 25.** *(Cartesian product) Let $\widetilde{G}_1(V_1, \widetilde{E}_1)$ and $\widetilde{G}_2(V_2, \widetilde{E}_2)$ be two fuzzy graphs, $\mu_1$ and $\mu_2$ be the membership functions of $\widetilde{E}_1$ and $\widetilde{E}_2$, respectively. The Cartesian product of two fuzzy graphs $\widetilde{G}_1$ and $\widetilde{G}_2$ is a fuzzy graph $\widetilde{G}(V, \widetilde{E}) = \widetilde{G}_1 \times \widetilde{G}_2$, where $V = V_1 \times V_2$, and $\widetilde{E} = \{(u_1v, v_1v) | v \in V_2, (u_1, v_1) \in \widetilde{E}_1\} \cup \{(uu_2, uv_2) | u \in V_1, (u_2, v_2) \in \widetilde{E}_2\}$,*

$$\mu(u_1v, v_1v) = \mu(u_1, v_1), \text{ for all } v \in V_2, \text{ for all } (u_1, v_1) \in \widetilde{E}_1,$$
$$\mu(uu_2, uv_2) = \mu(u_2, v_2), \text{ for all } u \in V_1, \text{ for all } (u_2, v_2) \in \widetilde{E}_2.$$

**Example 3.** *Consider two fuzzy graphs $\widetilde{G}_1(V_1, \widetilde{E}_1)$ and $\widetilde{G}_2(V_2, \widetilde{E}_2)$ with the underlying graph $G_1^*(V_1, E_1^*)$ and $G_2^*(V_2, E_2^*)$, respectively, such that $V_1 = \{u_1, u_2, u_3\}$, $\widetilde{E}_1 = \{(u_1, u_2), (u_2, u_3)\}$ and $V_2 = \{v_1, v_2, v_3\}$, $\widetilde{E}_2 = \{(v_1, v_2), (v_2, v_3)\}$, the $\mu_i$ corresponding to $\widetilde{G}_1$ and $\widetilde{G}_2$ are shown as follows:*

$$\mu_1 = \begin{bmatrix} - & 0.4 & 0 \\ 0.4 & - & 0.8 \\ 0 & 0.8 & - \end{bmatrix}, \qquad \mu_2 = \begin{bmatrix} - & 0.6 & 0 \\ 0.6 & - & 0.7 \\ 0 & 0.7 & - \end{bmatrix}.$$

*Then the direct product $\widetilde{G}_1 \sqcap \widetilde{G}_2$, the semiproduct $\widetilde{G}_1 \Diamond \widetilde{G}_2$, the strong product $\widetilde{G}_1 \otimes \widetilde{G}_2$ and the Cartesian product $\widetilde{G}_1 \times \widetilde{G}_2$ of above two fuzzy graphs $\widetilde{G}_1$ and $\widetilde{G}_2$ as shown in Figure 3a–d, respectively.*

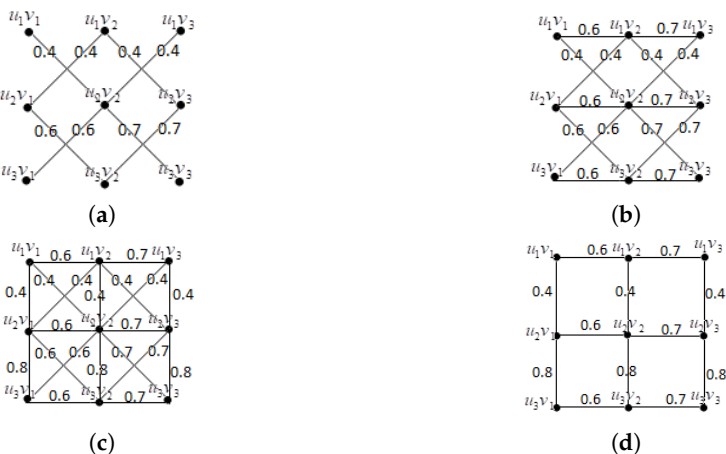

**Figure 3.** (a) $\widetilde{G}_1 \sqcap \widetilde{G}_2$, (b) $\widetilde{G}_1 \Diamond \widetilde{G}_2$, (c) $\widetilde{G}_1 \otimes \widetilde{G}_2$, and (d) $\widetilde{G}_1 \times \widetilde{G}_2$.

## 3. Chromatic Number of Fuzzy Graphs

**Lemma 1.** *Let $\widetilde{G}_1(V_1, \widetilde{E}_1)$ and $\widetilde{G}_2(V_2, \widetilde{E}_2)$ be two fuzzy graphs, $G_1^*$ and $G_2^*$ be the underlying graphs of $\widetilde{G}_1$ and $\widetilde{G}_2$, respectively. If $\widetilde{G}_1 \cong \widetilde{G}_2$ then $G_1^* \cong G_2^*$.*

**Proof.** Suppose that $\widetilde{G}_1 \cong \widetilde{G}_2$, and then there exists a bijection $h : V_1 \to V_2$ such that $\mu_1(u,v) = \mu_2(h(u), h(v))$ for all $u, v \in V_1$. It is obvious that there exists a bijection $h' : V_1 \to V_2$ such that $(u,v) = (h'(u), h'(v))$ for all $u, v \in V_1$, i.e., $G_1^* \cong G_2^*$. $\square$

**Lemma 2.** *For two fuzzy graphs $\widetilde{G}_1(V_1, \widetilde{E}_1)$ and $\widetilde{G}_2(V_2, \widetilde{E}_2)$, if $\widetilde{G}_1 \cong \widetilde{G}_2$ then $\chi(\widetilde{G}_1) = \chi(\widetilde{G}_2)$.*

**Proof.** Obviously, this lemma can be obtained from Lemma 1, and the definition of the chromatic number of fuzzy graphs. $\square$

**Theorem 1.** *Let $\widetilde{G}(V, \widetilde{E})$ be a self-complementary fuzzy graph, and the chromatic number of fuzzy graph $\bar{\bar{\widetilde{G}}}$ is the same with the fuzzy graph $\widetilde{G}$.*

**Proof.** By the definition of the self-complementary fuzzy graph, it is obvious that $\widetilde{G} \cong \bar{\bar{\widetilde{G}}}$. Furthermore, based on Lemma 2, it is obvious that $\chi(\widetilde{G}) = \chi(\bar{\bar{\widetilde{G}}})$. $\square$

**Lemma 3.** *Let $\widetilde{G}_1(V_1, \widetilde{E}_1)$ and $\widetilde{G}_2(V_2, \widetilde{E}_2)$ be two fuzzy graphs. If $\widetilde{G}_1 \subseteq \widetilde{G}_2$ then $\chi(\widetilde{G}_1) \leq \chi(\widetilde{G}_2)$.*

**Proof.** It is clear that $\chi(\widetilde{G}_1) \leq \chi(\widetilde{G}_2)$ if $\widetilde{G}_1 \subseteq \widetilde{G}_2$. $\square$

**Corollary 1.** *For two fuzzy graphs $\widetilde{G}_1(V_1, \widetilde{E}_1)$ and $\widetilde{G}_2(V_2, \widetilde{E}_2)$. If $\widetilde{G}_1 \subseteq \widetilde{G}_2$, then $\chi(\widetilde{G}_1 \cap \widetilde{G}_2) = \chi(\widetilde{G}_1)$.*

**Theorem 2.** *Let $\widetilde{G}(V, \widetilde{E})$ be a fuzzy graph. If $A = \{\alpha_1, \alpha_2, \cdots, \alpha_n\}$ is the fundamental set of $\widetilde{G}$, then for $\alpha_i, \alpha_j \in A$ such that $\alpha_i \leq \alpha_j$, we have $\chi(\widetilde{G}_{\alpha_j}) \leq \chi(\widetilde{G}_{\alpha_i})$.*

**Proof.** For the fuzzy graph $\widetilde{G}(V, \widetilde{E})$, if $0 \leq \alpha_i \leq \alpha_j \leq 1$, then $G_{\alpha_i}(V, E_{\alpha_i})$ is a crisp graph, where $E_{\alpha_i} = \{(u,v)|\mu(u,v) \geq \alpha_i, u, v \in V\}$. Meanwhile, $G_{\alpha_j}(V, E_{\alpha_j})$ is a crisp graph, where $E_{\alpha_j} = \{(u,v)|\mu(u,v) \geq \alpha_j, u, v \in V\}$.

Let $(u,v) \in E_{\alpha_j}$, then $\mu(u,v) \geq \alpha_j \geq \alpha_i$, i.e., $(u,v) \in E_{\alpha_i}$. Therefore, $G_{\alpha_j} \subseteq G_{\alpha_i}$, and by Lemma 3, $\chi(\widetilde{G}_{\alpha_j}) \leq \chi(\widetilde{G}_{\alpha_i})$. $\square$

**Theorem 3.** *Let $\widetilde{G}_1(V_1, \widetilde{E}_1)$ and $\widetilde{G}_2(V_2, \widetilde{E}_2)$ be two fuzzy graphs, the chromatic numbers of $\widetilde{G}_1$ and $\widetilde{G}_2$ be $\chi(\widetilde{G}_1)$ and $\chi(\widetilde{G}_2)$, respectively. If fuzzy graph $\widetilde{G}(V, \widetilde{E})$ is the union of two fuzzy graphs $\widetilde{G}_1$ and $\widetilde{G}_2$, then the chromatic number of $\widetilde{G}$ satisfies $\max\{\chi(\widetilde{G}_1), \chi(\widetilde{G}_2)\} \leq \chi(\widetilde{G}) \leq \chi(\widetilde{G}_1) + \chi(\widetilde{G}_2)$.*

**Proof.** We will construct the union of fuzzy graphs $\widetilde{G}_1$ and $\widetilde{G}_2$. We consider the following two cases.

Case 1. $V_1 \cap V_2 = \emptyset$. Assume that $\mu$, $\mu_1$ and $\mu_2$ are the membership functions of $\widetilde{E}$, $\widetilde{E}_1$ and $\widetilde{E}_2$, respectively. Because $V_1 \cap V_2 = \emptyset$, it is clear that $\widetilde{G} = \widetilde{G}_1 \cup \widetilde{G}_2 = (V_1 \cup V_2, \widetilde{E})$ with $\mu(u, v) > 0$ for all $u, v \in V_1 \cup V_2$ if and only if $\mu_1(u, v) > 0$ or $\mu_2(u, v) > 0$. For each $\alpha_i$ in the fundamental set of $\widetilde{G}$, either $\alpha_i$ belongs to the fundamental set of $\widetilde{G}_1$ or to the fundamental set of $\widetilde{G}_2$. Hence, the minimum value of $\alpha_i$ in the fundamental set of $\widetilde{G}$ is the minimum among that in the fundamental sets of $\widetilde{G}_1$ and $\widetilde{G}_2$. Let it be $\alpha_k$, the minimum value of $\alpha_i$ in the fundamental set of $\widetilde{G}_1$. Then $(\widetilde{G}_1 \cup \widetilde{G}_2)_{\alpha_k}$ has all the adjacencies of $\widetilde{G}_1$ and $\widetilde{G}_2$. By Definition 4, $\chi(\widetilde{G}_1 \cup \widetilde{G}_2) = \chi(\widetilde{G}_1 \cup \widetilde{G}_2)_{\alpha_k} = \max\{\chi(\widetilde{G}_1), \chi(\widetilde{G}_2)\}$.

Case 2. $V_1 \cap V_2 \neq \emptyset$.

In this case, there are two possibilities as follows.

(i) There do not form new cycles in fuzzy graphs $\widetilde{G}(V, \widetilde{E})$. It is obvious that $\chi(\widetilde{G}) = \max\{\chi(\widetilde{G}_1), \chi(\widetilde{G}_2)\}$.

(ii) There are new cycles formed in the fuzzy graph $\widetilde{G}(V, \widetilde{E})$. In order to satisfy $f(u) \neq f(v)$, new colors may be required in $\widetilde{G}$. It is obvious that $\max\{\chi(\widetilde{G}_1), \chi(\widetilde{G}_2)\} \leq \chi(\widetilde{G}) \leq \chi(\widetilde{G}_1) + \chi(\widetilde{G}_2)$.

Combining the above results, we obtain that $\max\{\chi(\widetilde{G}_1), \chi(\widetilde{G}_2)\} \leq \chi(\widetilde{G}) \leq \chi(\widetilde{G}_1) + \chi(\widetilde{G}_2)$. □

**Corollary 2.** *For two fuzzy graphs $\widetilde{G}_1(V_1, \widetilde{E}_1)$ and $\widetilde{G}_2(V_2, \widetilde{E}_2)$. If $\widetilde{G}_1 \subseteq \widetilde{G}_2$, then $\chi(\widetilde{G}_1 \cup \widetilde{G}_2) = \chi(\widetilde{G}_2)$.*

**Theorem 4.** *Let $\widetilde{G}_1(V_1, \widetilde{E}_1)$ and $\widetilde{G}_2(V_2, \widetilde{E}_2)$ be two fuzzy graphs, the chromatic numbers of $\widetilde{G}_1$ and $\widetilde{G}_2$ be $\chi(\widetilde{G}_1)$ and $\chi(\widetilde{G}_2)$, respectively. If $\widetilde{G}(V, \widetilde{E})$ is the difference of two fuzzy graphs $\widetilde{G}_1$ and $\widetilde{G}_2$, then the chromatic number of $\widetilde{G}$ satisfies $\chi(\widetilde{G}) \leq \chi(\widetilde{G}_1)$.*

**Proof.** By the definition of the difference between fuzzy graphs, we have $\widetilde{G} = \widetilde{G}_1 - \widetilde{G}_2 \subseteq \widetilde{G}_1$. Moreover, based on Lemma 3, it is obvious that $\chi(\widetilde{G}) = \chi(\widetilde{G}_1 - \widetilde{G}_2) \leq \chi(\widetilde{G}_1)$. □

**Theorem 5.** *Let $\widetilde{G}_1(V_1, \widetilde{E}_1)$ and $\widetilde{G}_2(V_2, \widetilde{E}_2)$ be two fuzzy graphs, the chromatic numbers of $\widetilde{G}_1$ and $\widetilde{G}_2$ be $\chi(\widetilde{G}_1)$ and $\chi(\widetilde{G}_2)$, respectively. If $\widetilde{G}(V, \widetilde{E})$ is the cap of two fuzzy graphs $\widetilde{G}_1$ and $\widetilde{G}_2$, then the chromatic number of $\widetilde{G}$ satisfies $\chi(\widetilde{G}) \leq \min\{\chi(\widetilde{G}_1), \chi(\widetilde{G}_2)\}$.*

**Proof.** By the definition of the cap of two fuzzy graphs, $\widetilde{G} = \widetilde{G}_1 \cap \widetilde{G}_2$, $\widetilde{E} = \widetilde{E}_1 \cap \widetilde{E}_2$. Also, $\widetilde{G} \subseteq \widetilde{G}_1$ and $\widetilde{G} \subseteq \widetilde{G}_2$. By Lemma 3, it is clear that $\chi(\widetilde{G}) \leq \min\{\chi(\widetilde{G}_1), \chi(\widetilde{G}_2)\}$. □

**Theorem 6.** *Let $\widetilde{G}_1(V_1, \widetilde{E}_1)$ and $\widetilde{G}_2(V_2, \widetilde{E}_2)$ be two fuzzy graphs, the chromatic numbers of $\widetilde{G}_1$ and $\widetilde{G}_2$ be $\chi(\widetilde{G}_1)$ and $\chi(\widetilde{G}_2)$, respectively. If $\widetilde{G}(V, \widetilde{E})$ is the ring sum of two fuzzy graphs $\widetilde{G}_1$ and $\widetilde{G}_2$, then the chromatic number of $\widetilde{G}$ satisfies $\chi(\widetilde{G}) \leq \chi(\widetilde{G}_1) + \chi(\widetilde{G}_2)$.*

**Proof.** By the definition of the ring sum in fuzzy graphs, we know $\widetilde{G}_1 \oplus \widetilde{G}_2 = (\widetilde{G}_1 \cup \widetilde{G}_2) - (\widetilde{G}_1 \cap \widetilde{G}_2) \subseteq \widetilde{G}_1 \cup \widetilde{G}_2$. Based on the proof of Theorem 3 and Lemma 3, there are $\chi(\widetilde{G}) = \chi(\widetilde{G}_1 \oplus \widetilde{G}_2) \leq \chi(\widetilde{G}_1 \cup \widetilde{G}_2) \leq \chi(\widetilde{G}_1) + \chi(\widetilde{G}_2)$. □

**Theorem 7.** *Let $\widetilde{G}_1(V_1, \widetilde{E}_1)$ and $\widetilde{G}_2(V_2, \widetilde{E}_2)$ be two fuzzy graphs, the chromatic numbers of $\widetilde{G}_1$ and $\widetilde{G}_2$ be $\chi(\widetilde{G}_1)$ and $\chi(\widetilde{G}_2)$, respectively. If $\widetilde{G}(V, \widetilde{E})$ is the join of two fuzzy graphs $\widetilde{G}_1$ and $\widetilde{G}_2$, then the chromatic number of $\widetilde{G}$ satisfies $\chi(\widetilde{G}) = \chi(\widetilde{G}_1 + \widetilde{G}_2) \leq n$ where $n = |V_1| + |V_2|$. Equality holds if $G_1^*$ and $G_2^*$ are complete graphs, where $G_1^*$ and $G_2^*$ are the underlying graphs of $\widetilde{G}_1$ and $\widetilde{G}_2$, respectively.*

**Proof.** By the definition of join in fuzzy graphs, the underlying graph $G^*$ of join graph $\widetilde{G}$ becomes a complete graph if and only if both $G_1^*$ and $G_2^*$ are complete graphs. □

**Remark 3.** *Let $\widetilde{G}_1$ and $\widetilde{G}_2$ be two fuzzy graphs, the fuzzy graph $\widetilde{G}$ is the join of $\widetilde{G}_1$ and $\widetilde{G}_2$, then $\widetilde{G}$ is a fuzzy complete graph if $\widetilde{G}_1$ and $\widetilde{G}_2$ are both fuzzy complete graphs.*

**Theorem 8.** *For two fuzzy graphs $\widetilde{G}_1(V_1, \widetilde{E}_1)$ and $\widetilde{G}_2(V_2, \widetilde{E}_2)$, the chromatic number of the direct product $\chi(\widetilde{G}_1 \sqcap \widetilde{G}_2) = \min\{\chi(\widetilde{G}_1), \chi(\widetilde{G}_2)\}$.*

**Proof.** By the definition of direct product of fuzzy graphs, any two vertices are adjacent if and only if the corresponding vertices in $\widetilde{G}_1$ and $\widetilde{G}_2$ are adjacent. Because the chromatic number depends on adjacency, the number of colors required to color the vertices of $\widetilde{G}_1 \sqcap \widetilde{G}_2$ is the minimum of $\chi(\widetilde{G}_1)$ and $\chi(\widetilde{G}_2)$. □

**Theorem 9.** *For two fuzzy graphs $\widetilde{G}_1(V_1, \widetilde{E}_1)$ and $\widetilde{G}_2(V_2, \widetilde{E}_2)$, the chromatic number of the semiproduct $\chi(\widetilde{G}_1 \lozenge \widetilde{G}_2) = |V_C|$. Where C is a maximal clique of $(\widetilde{G}_1 \lozenge \widetilde{G}_2)^*$.*

**Proof.** By the definition of semiproduct, the largest complete subgraph of $\widetilde{G}$ is the graph with corresponding adjacent vertices $u_i v_j$ and $u_i v_k$, where $u_i \in V_1$ and $(v_j, v_k) \in \widetilde{E}_2$, i.e., the largest complete subgraph is the graph with edges of the form $(u_1 v_j, u_1 v_k)$ with $u_1$ as a vertex of maximum degree in $V_1$ and edges of the form $(u_1 v_j, u_m v_k)$, where $(u_1, u_m) \in \widetilde{E}_1$ and $(v_j, v_k) \in \widetilde{E}_2$. Hence the chromatic number of the semiproduct is the chromatic number corresponding to this largest complete subgraph. □

**Theorem 10.** *For two fuzzy graphs $\widetilde{G}_1(V_1, \widetilde{E}_1)$ with the membership function $\mu_1$ and $\widetilde{G}_2(V_2, \widetilde{E}_2)$ with the membership function $\mu_2$. If $\mu_1(u_1, u_2) > 0$ and $\mu_2(v_1, v_2) > 0$, for all $u_1, u_2 \in V_1$ and $v_1, v_2 \in V_2$, the chromatic number of the strong product $\chi(\widetilde{G}_1 \otimes \widetilde{G}_2) = n_1 n_2$, where $n_1 = |V_1|$ and $n_2 = |V_2|$.*

**Proof.** By the definition of strong product, if $\mu_1(u_1, u_2) > 0$ and $\mu_2(v_1, v_2) > 0$, for all $\mu_1, \mu_2 \in V_1$ and $v_1, v_2 \in V_2$, then $(\widetilde{G}_1 \otimes \widetilde{G}_2)^*$ is a complete graph with $n_1 n_2$ vertices. Hence, $\chi(\widetilde{G}_1 \otimes \widetilde{G}_2) = n_1 n_2$. □

**Theorem 11.** *For two fuzzy graphs $\widetilde{G}_1(V_1, \widetilde{E}_1)$ and $\widetilde{G}_2(V_2, \widetilde{E}_2)$, the chromatic number of the Cartesian product $\max\{\chi(\widetilde{G}_1), \chi(\widetilde{G}_2)\} \leq \chi(\widetilde{G}_1 \times \widetilde{G}_2) \leq n_2 \chi(\widetilde{G}_1) + n_1 \chi(\widetilde{G}_2)$, where $n_1 = |V_1|$ and $n_2 = |V_2|$.*

**Proof.** The Cartesian product graph $\widetilde{G}_1 \times \widetilde{G}_2$ decomposes into $n_1$ copies of $\widetilde{G}_2$ and $n_2$ copies of $\widetilde{G}_1$, where $|V_1| = n_1$ and $|V_2| = n_2$. By the definition of Cartesian product, $\widetilde{G}_1 \times \widetilde{G}_2$ has two types of edges: those with vertices that have the same first coordinate, and those with vertices that have the same second coordinate. The edges joining vertices with a given value of the first coordinate form a copy of $\widetilde{G}_2$, so the edges of the first type form $n_1 \widetilde{G}_2$. Similarly, the edges of the second type form $n_2 \widetilde{G}_1$, and the union is as follows:

$$(\underbrace{\widetilde{G}_1 \cup \widetilde{G}_1 \cdots \cup \widetilde{G}_1}_{n_2}) \cup (\underbrace{\widetilde{G}_2 \cup \widetilde{G}_2 \cdots \cup \widetilde{G}_2}_{n_1}).$$

Based on the proof of Theorem 3, it is obvious that $\max\{\chi(\widetilde{G}_1), \chi(\widetilde{G}_2)\} \leq \chi(\widetilde{G}_1 \times \widetilde{G}_2) \leq n_2 \chi(\widetilde{G}_1) + n_1 \chi(\widetilde{G}_2)$. □

## 4. Application of Chromatic Number of Fuzzy Graphs

In this section, we describe a couple of example applications of the chromatic number of fuzzy graphs.

### 4.1. The Examination Problem

The examination scheduling problem consists of assigning a number of exams to a number of slots within the examination period, taking into account that students cannot

take more than one exam at the same time. This problem can be modeled as a classical coloring problem. Each exam is a vertex of the graph, and the edges link those incompatible exams in the sense that at least one student shares both of them. Each color represents a slot, and all equally colored vertices are assigned to the same slot. The minimum chromatic number has the same property of minimizing the overall exam period.

Now, we take into account the difficulty and the implied specialties or profiles and other factors of the examination subjects. The incompatibility between the exams is valued in the following way: $n$ if they are compatible, and $l$, $m$ and $h$ if the incompatibility degree is, respectively, low, medium, and high. The incompatibility degree can be interpreted in the following way: any compatible exams can share the same slot; if the incompatibility degree between exams $i$ and $j$ is $l$, $m$, or $h$, then both exams must be scheduled, leaving at least 1, 2, or 3 slots between them, respectively (to ensure that students have relatively sufficient review time). Then, this problem can be modeled by means of fuzzy graphs.

**Example 4.** *Suppose there are eight exams* $\{A, B, C, D, E, F, G, H\}$*. The model of this examination scheduling problem is a fuzzy graph* $\widetilde{G}(V, \mu)$*, where* $V = \{A, B, C, D, E, F, G, H\}$,

$$
\mu = \begin{bmatrix}
- & l & m & n & n & l & n & n \\
l & - & m & n & n & n & n & n \\
m & m & - & l & n & h & n & n \\
n & n & l & - & m & n & n & l \\
n & n & n & m & - & l & n & n \\
l & n & h & n & l & - & m & n \\
n & n & n & n & n & m & - & l \\
n & n & n & n & n & n & l & -
\end{bmatrix}.
$$

*The corresponding fuzzy graphs is depicted in Figure 4.*

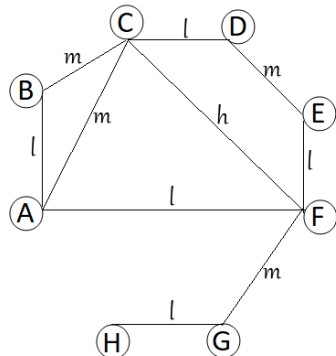

**Figure 4.** Fuzzy graph $\widetilde{G}$ of Example 4 (*A*–*H* represent different exams respectively).

The problem is to schedule the eight exams with these constraints and minimizing the overall exam period. For the above data, the minimum number of slots is four and an optimum schedule (in other words we need 4 colours to dye the vertices in this fuzzy graph) by Table 1.

**Table 1.** Examination scheduling.

| Slots | 1 | 2 | 3 | 4 |
|---|---|---|---|---|
| Scheduled exams | H,F,D,B | A | E,G | C |

*4.2. The Traffic Light System Problem*

Let us consider a four-way crossroad at two intersections, as illustrated in Figure 5. There are four directions on the left side intersection, i.e., *A* on the north, *B* on the south, *C* on the west, and *D* on the east. Moreover, there are four directions at the right side of the intersection, i.e., *E*, *F*, *C*, and *D*. Thus, there are traffic movements in different directions, i.e., *AB*, *AD*, *BA*, *BC*, *CD*, *CA*, *DC*, *DB*, *ED*, *DF*, and *EF*. Certain routes, such as *AD* and *BC*, *CD* and *DC*, and *AD* and *CD*, are compatible with one another, whereas others, such as *AB* and *BC*, *EF* and *CD*, are not.

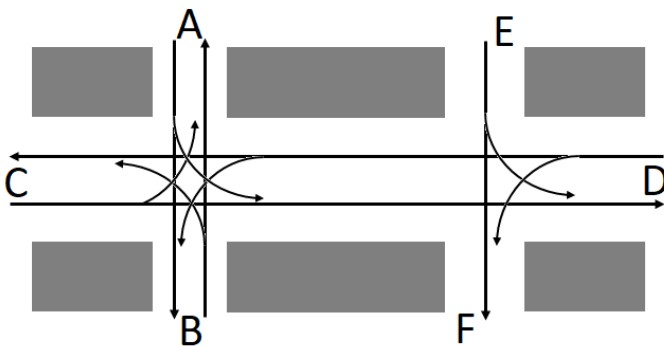

**Figure 5.** A traffic light system on two intersections (*A–F* represent one direction of the intersection respectively).

A vertex represents the movement of traffic from one direction to another. Two vertices that represent incompatible traffic movements should be connected with an edge. The traffic light system will be modified by the vertex coloring of fuzzy graphs. The minimum chromatic number has the same property of minimizing the traffic light phases.

One can represent the two signaled intersections through the union of two fuzzy graphs. A membership degree of a fuzzy edge represents the incompatibility of two vertices, which is a possibility for traffic incidents to occur between vehicles from both traffic movements.

The first stage in research is to divide the data on traffic flows into three groups of fuzzy sets, i.e., low, medium, or high flows, as shown in Table 2.

**Table 2.** Data of traffic flows (vehicles/h).

|  | *AB* | *AD* | *BA* | *BC* | *CD* | *CA* | *DC* | *DB* | *ED* | *DF* | *EF* |
|---|---|---|---|---|---|---|---|---|---|---|---|
| *n* | 536 | 350 | 476 | 425 | 582 | 396 | 678 | 313 | 275 | 317 | 465 |
| Degree (*l*) |  |  |  |  |  |  |  | 0.10 | 0.33 | 0.08 |  |
| Degree (*m*) | 0.43 | 0.33 | 0.83 | 0.83 | 0.12 | 0.64 |  | 0.09 |  | 0.11 | 0.9 |
| Degree (*h*) |  |  |  |  | 0.07 |  | 0.47 |  |  |  |  |

The symbols *l*, *m*, and *h* mean low, medium, and high, respectively. The notation *n* denotes the number of vehicles per hour (vehicles/h) involved in a traffic movement. A trapezoidal membership function at intervals of $[300, 600]$ is used to present the fuzzy set of low traffic flow, a triangular membership function at intervals of $[300, 600]$ is used to present the fuzzy set of medium traffic flow, and a trapezoidal membership function at intervals of $[570, 900]$ is used to present the fuzzy set of high traffic flow. Figure 6 depicts the fuzzification of traffic flow data in Table 2. Displaying the membership degree of an edge is helpful in order to understand the incompatibility of two traffic flows.

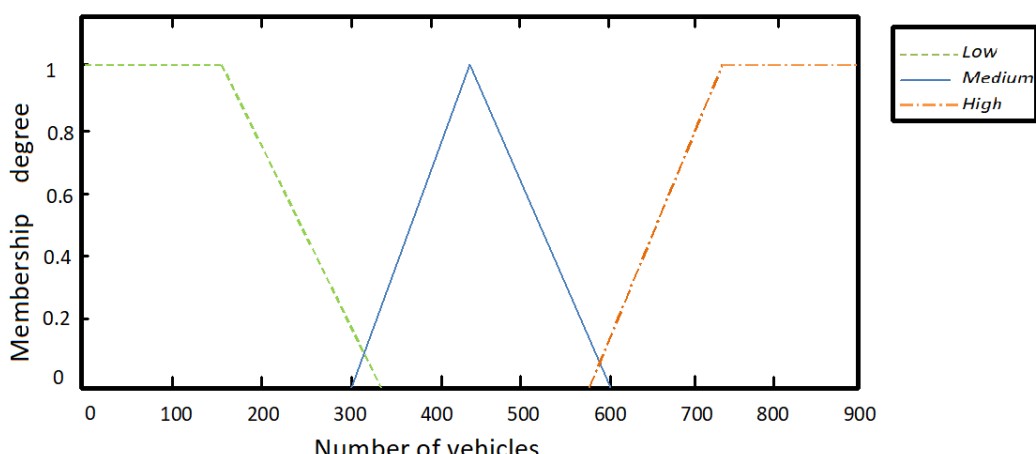

**Figure 6.** Membership functions of low, medium, and high traffic flows.

The second step is to determine the degrees of membership of all edges by using a rule as follows.

(i) If traffic movements $AB$ and $CD$ are incompatible, then there is an edge $(AB, CD)$. In addition, we choose a maximum value of membership degree of $AB$ and $CD$ to determine the membership degree of the fuzzy edge $(AB, CD)$.
(ii) If the traffic movements $AB$ and $CD$ are compatible, which means they can move at the same time, then there is no $(AB, CD)$ edge. This indicates that the fuzzy edge's membership degree is 0.

Therefore, traffic flows on the left and right intersection in Figure 5 can be modelled as two fuzzy graphs $\widetilde{G}_1(V_1, \widetilde{E}_1)$ and $\widetilde{G}_2(V_2, \widetilde{E}_2)$ based on the given rules, respectively, where the vertex sets are $V_1 = \{AB, AD, BA, BC, CD, CA, DC, DB\}$ and $V_2 = \{DC, ED, DF, CD, EF, AD, DB\}$.

The membership degrees of edges in fuzzy edge set $\widetilde{E}$ is presented in Table 3. One can obtain the fundamental set $A = \{0.12, 0.33, 0.43, 0.47, 0.64, 0.83,$ and $0.9\}$ of fuzzy graph $\widetilde{G}$ from Table 3. It can be known that routes $DC$ and $EF$ are more incompatible than routes $DC$ and $CA$. The fuzzy graphs $\widetilde{G}_1(V_1, \widetilde{E}_1)$ and $\widetilde{G}_2(V_2, \widetilde{E}_2)$ are shown as Figure 7.

**Table 3.** Fuzzy edge sets.

| Movements | AB | AD | BA | BC | CD | CA | DC | DB | ED | DF | EF |
|-----------|-----|-----|-----|-----|-----|-----|-----|-----|-----|-----|-----|
| AB | - | - | - | 0.83 | 0.43 | 0.64 | 0.47 | - | - | - | - |
| AD | - | - | 0.83 | - | - | 0.64 | 0.47 | 0.33 | - | - | 0.9 |
| BA | - | 0.83 | - | - | 0.83 | - | 0.83 | 0.83 | - | - | - |
| BC | 0.83 | - | - | - | 0.83 | 0.83 | - | 0.83 | - | - | - |
| CD | 0.43 | - | 0.83 | 0.83 | - | - | - | 0.12 | - | 0.12 | 0.9 |
| CA | 0.64 | 0.64 | - | 0.83 | - | - | 0.64 | 0.64 | - | - | - |
| DC | 0.47 | 0.47 | 0.83 | - | - | 0.64 | - | - | 0.47 | - | 0.9 |
| DB | - | 0.33 | 0.83 | 0.83 | 0.12 | 0.64 | - | - | - | - | 0.9 |
| ED | - | - | - | - | - | - | 0.47 | - | - | 0.33 | - |
| DF | - | - | - | - | 0.12 | - | - | - | 0.33 | - | - |
| EF | - | 0.9 | - | - | 0.9 | - | 0.9 | 0.9 | - | - | - |

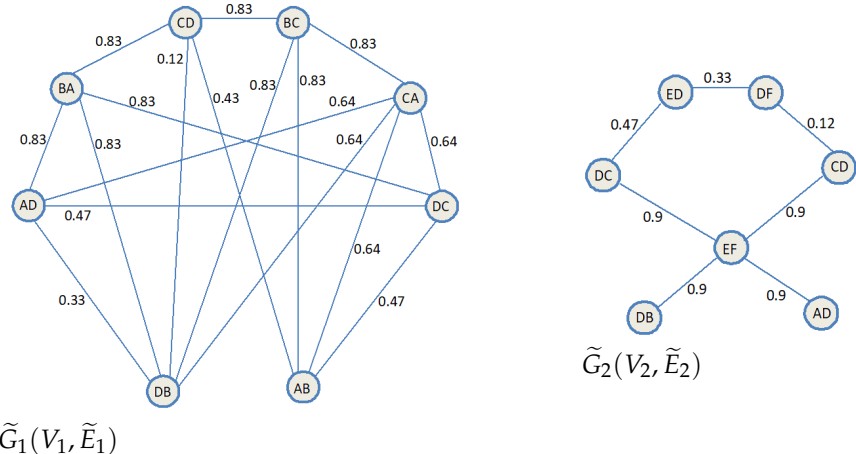

Figure 7. Fuzzy graph models for traffic light system in Figure 5.

There exist some methods of colouring in fuzzy graphs. It is not difficult to get chromatic numbers of fuzzy graphs $\widetilde{G}_1$ and $\widetilde{G}_2$ by using any chromatic algorithm. The chromatic numbers of $\widetilde{G}_1$ and $\widetilde{G}_2$ are as follows: $\chi(\widetilde{G}_1) = 4$ and $\chi(\widetilde{G}_2) = 3$.

Furthermore, two intersections in Figure 5 can be modelled into an integrated traffic light system by using a union of fuzzy graphs $\widetilde{G} = \widetilde{G}_1 \cup \widetilde{G}_2$ as illustrated in Figure 8. According to Definition 11 and Theorem 3, we get chromatic number of union in Figure 7 as follows: $\chi(\widetilde{G}) = \max\{\chi(G_{\alpha_i}) | \alpha_i \in A\} = \chi(\widetilde{G}_1 \cup \widetilde{G}_2) = 4$.

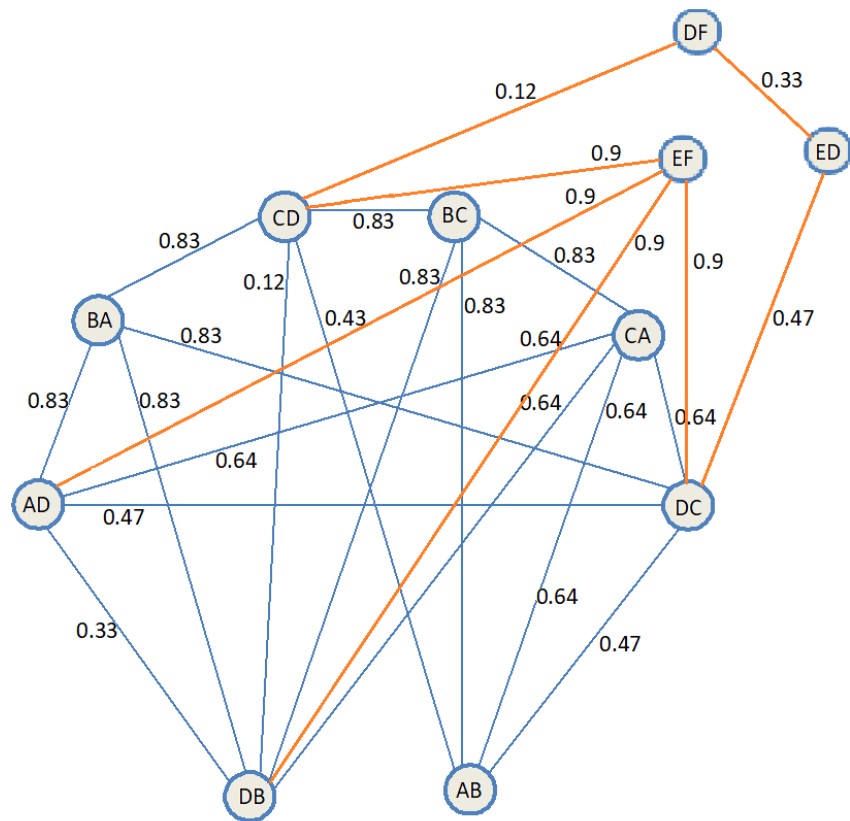

Figure 8. The model of union of fuzzy graphs in Figure 7.

The chromatic number of crisp graphs $\chi(G_{\alpha_i}) = k$ can be interpreted as follows: the number $k$ represents the number of phases needed on the integrated system and degree

of membership $\alpha_i$ represents the possibility that there may occur traffic incidents when we use $k$ phases.

Based on chromatic number $k$ of $\alpha_i$-cut graph of fuzzy graph $\widetilde{G}(V, \widetilde{E})$, traffic flows in Figure 5 could be arranged in particular patterns as shown in Table 4. For higher values of $\alpha_i$ (i.e., the higher degree of incompatibility), there are fewer colors needed and less safety. For example, when we make use of $k = 3$ phases, then 0.64 degree of incompatibility is reached. Traffic flows allowed to move simultaneously (i.e., get the green light) on the first phase are $AD$, $DB$, $AB$, $CD$ and $DC$, $BC$, $BA$, $EF$ on the second phase, $CA$, $DF$ and $ED$ on the third phase.

Finally, one can obtain an optimal arrangement. The optimality is reached when maximum safety level is attained (i.e., the minimum degree of incompatibility) whereas the number of phases is minimum (i.e., minimum cycle time of traffic light). When $k = 4$, the objective is reached, traffic flows allowed to move simultaneously on the first phase are $AD$, $CD$ and $ED$, $BA$, $CA$, $DF$, and $EF$ on the second phase, $BC$ and $DC$ on the third phase, and $DB$ and $AB$ on the fourth phase. Only in this way can the maximum avoid traffic incidents.

**Table 4.** Traffic light arrangements by using $k$ phases.

| $\alpha_i$ | $k$ | Arrangements (Partitions of Vertex Set $V$) |
|---|---|---|
| 0.12 | 4 | $\{AD, CD, ED\}, \{BA, CA, DF, EF\}, \{BC, DC, \}, \{DB, AB\}$ |
| 0.33 | 4 | $\{AD, CD, ED\}, \{BA, CA, DF, EF\}, \{BC, DC, \}, \{DB, AB\}$ |
| 0.43 | 4 | $\{AD, CD, ED\}, \{BA, CA, DF, EF\}, \{BC, DC, \}, \{DB, AB\}$ |
| 0.47 | 3 | $\{AD, DB, AB, EF, ED\}, \{BA, BC, DC\}, \{DF, CD, CA\}$ |
| 0.64 | 3 | $\{AD, DB, AB, CD, DC\}, \{BC, BA, EF\}, \{CA, DF, ED\}$ |
| 0.83 | 2 | $\{AD, CD, DC, DB, AB\}, \{BA, BC, DF, ED, EF\}$ |
| 0.9 | 2 | $\{AD, CD, DC, DB, AB\}, \{BA, BC, DF, ED, EF\}$ |

## 5. Conclusions

In this paper, the concepts of complement, cap, join, difference, ring sum, and various categories of products of fuzzy graphs with a crisp vertex set and a fuzzy edge set are defined, and the corresponding chromatic number or upper bound is obtained. Furthermore, the chromatic number is used to solve the problems of the exam schedule and traffic light system. Finally, we get the optimal arrangement by finding the minimum chromatic number of a fuzzy graph.

In our upcoming research, we will define the fuzzy acyclic coloring of fuzzy graphs. In addition, we will verify some properties of fuzzy acyclic chromatic numbers of fuzzy graphs.

**Author Contributions:** Z.G. and J.Z. completed the study together. J.Z. wrote the manuscript, Z.G. checked the proofs process and verified the calculation. All authors have read and agreed to the published version of the manuscript.

**Funding:** This research was supported by the National Natural Science Foundation of China (No.12061067).

**Institutional Review Board Statement:** Not applicable.

**Informed Consent Statement:** Not applicable.

**Data Availability Statement:** Not applicable.

**Acknowledgments:** The authors would like to thank the referees for providing very helpful comments and suggestions.

**Conflicts of Interest:** The authors declare no conflict of interest.

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
