# Peer review of "Chromatic Number of Fuzzy Graphs: Operations, Fuzzy Graph Coloring, and Applications"

_axioms, doi:10.3390/axioms11120697_

Round 1

Reviewer 1 Report

See attached pdf file.

Author Response

refer to the submitted file: Revision Note of axioms-2039421 Revised

Reviewer 2 Report

I believe that the results are correct. However I do have some comments on the presentation.

On the definitions

Since you decided to define everything from the very basics, it seems that you need to do it more thoroughly. For example, it seems that terms like "fuzzy relation" and "type 1" have not been formally defined.

I am also a bit puzzled by some of the definitions. For instance, in the definition of k-colouring, do we really need the sets S_i to be non-empty? (I think most books on graph theory would not require that.) When you said $V \times V$ in the definition of fuzzy graphs, do you mean the edges are directed and loops are allowed? But it seems from the later discussions that this is not really the case. Some clarifications on these may be necessary.

On typography

There are a few issues with the typography. To name some examples:

-- "$k$-colouring" is typed as "$k-$colouring"

-- On several occasions mathematical expressions are not in math mode (Page 5 Line 11, Page 12 Table 4.1, etc.)

-- The notation for induced subgraph is typed as $$; it is much better to use something like $\langle P\rangle$.

On language

There seems to be some grammatical issues (or possibly typos) throughout. To name a couple of examples:

"The fuzzy graphs with crisp vertex sets and fuzzy edge sets, this type of fuzzy graphs are also known as the type 1 fuzzy graphs, were originally proposed by Kaufmann [22] in 1973."

The above sentence structure is not grammatically sound. Also in the following:

In this paper, we deal with fuzzy graphs G=(V,E) which denoted the incompatibility graph.

Again, there is something wrong, at least with the verb form.

The above are just two examples and there are quite a few more. I suggest that you check more carefully.

Author Response

refer to the submitted file Revision Note of axioms-2039421 Revised.

Reviewer 3 Report

The problem statement should be addressed in clear manner, now a weak intent appears to be described

The Definition 2.16 is unclear what kind of value belong or how it can be obtained for μ and ν

The Example 2.17 is nuclear how can be represented μ and ν. I suggest to authors to explain in good manner that. Also, it can be interesting to readers and practitioners to replicate the methods proposals.

The Example 2.27 still confuse need be explained in clear manner

The Definition 2.26 even need be clarified how the matrix is calculated, the readers can be interested in use the methodology proposed

For this case it is very confused to understand for the readers

Conclusion the future works still pending to include future works and doesn't explain if the objectives were to reach

Author Response

(The authors gave the same response as above.)

Round 2

Reviewer 1 Report

This new version improves the presentation of theory and results. However, my initial criticism concerning the lack of novelty and the paucity of new results remains.

Nevertheless, I'm not opposed any longer to the publication of this new version.